# A 4500-Year Tree-Ring Record of Extreme Climatic Events on the Yamal Peninsula

**Rashit Hantemirov** [1,2,†], **Liudmila Gorlanova** [2,†], **Varvara Bessonova** [1,2], **Ildar Hamzin** [1] and **Vladimir Kukarskih** [1,2,*]

1. School of Academic and Project Development, Ural Institute of Humanities, Ural Federal University, Ekaterinburg 620002, Russia; rashit@ipae.uran.ru (R.H.)
2. Institute of Plant and Animal Ecology, Ural Division of the Russian Academy of Sciences, Ekaterinburg 620144, Russia
* Correspondence: voloduke@mail.ru
† These authors contributed equally to this work.

**Abstract:** Based on the analysis of the frequency of anomalous anatomical structures in the wood of Siberian larch and Siberian spruce (frost rings, light rings, and false rings, as well as missing and narrow rings), we reconstructed a timeline of climatic extremes (summer frosts, sharp multiday decreases in air temperature during the growing season, and low average summer temperatures) in Yamal (Western Siberia) over the last 4500 years. In total, 229 years were determined to have experienced extreme events. The most significant temperature extremes were recorded in 2053, 1935, 1647, 1626, 1553, 1538, 1410, 1401, 982, 919, 883 BCE, 143, 404, 543, 640, 1209, 1440, 1453, 1466, 1481, 1601 and 1818 CE. These dates with extrema observed in Yamal corroborated with tree ring data from other regions and revealed several coincidences. That is, in these years, the observed extremes appeared to have been on a global rather than a regional scale. Moreover, these dates coincided with traces of large volcanic eruptions found in ice cores from Greenland and Antarctica, dated to approximately the same years. Therefore, the cause of the extreme summer cooling on a global scale, in most cases, can be linked to large volcanic eruptions.

**Keywords:** Yamal; tree rings; anomalous structures; Holocene; *Larix sibirica*; reconstruction

## 1. Introduction

Regular instrumental observations of climatic extremes usually do not exceed 100 years, owing to the relatively recent organization of meteorological stations. This hinders the identification of patterns of climate extremes on the century and millennial-scale. In its latest report, the Intergovernmental Panel on Climate Change [1] noted a lack of homogeneous data on long-term observations of climatic extremes which would allow comparisons to be made on the frequency and intensity of extremes in the past and modern eras. Therefore, information on climatic extremes reconstructed with indirect data seems highly valuable.

One of the most promising methods for reconstructing climatic extremes in the far past is based on the analysis of abnormal tissue and cell structures in tree rings [2–8]. This method makes it possible to reconstruct events that are accurate to the year, and in some cases, it is possible to pinpoint the event accurately within a range of one or two weeks. Reconstructions of events from up to several hundred years ago can be performed based on data from living trees [9–11]. Although these data on living trees span several centuries, it remains difficult to determine whether the recorded frequency of extreme events falls within the normal range of variability or is considered unusual. A big limitation regarding assessing changes in the frequency of extreme events is the lack of millennial-length records. The number of such records are extremely rare and can be counted on one hand [12–14]. The maximum length of a tree-ring anomaly series is a 5000-year span [15].

A large amount of well-preserved ancient (sub-fossil) wood from the Holocene age can be found in some subarctic areas. This makes it possible to extend tree-ring chronologies far back in time. Currently, a multi-millennial tree-ring chronology of more than 8500 years has been constructed on the Yamal Peninsula [16]. This chronology is based on the ring width of Siberian larch (*Larix sibirica* Ledeb.) growing in high latitudes in areas where trees are most sensitive to changes in the summer temperature [17]. In the process of constructing this chronology, cross-dating determined the dates of formation of annual rings in several thousand sub-fossil trees. This absolutely dated dendroclimatic material was used in this study.

Here, based on an analysis of anomalous anatomical structures in tree rings, we present one of the world's longest reconstructions of extreme temperature events during the growing season on the Yamal Peninsula over 4500 years. In this paper, we provide data on the types of anomalous structures in tree rings of larch and spruce, the frequency of their formation, and the change in frequency over time. We present a list of extreme years and compare them with similar data from remote regions in order to estimate the spatial extent of the extreme event. We also compared these data to records of volcanic deposition in ice cores to estimate the possible cause of the climatic extremes.

## 2. Materials and Methods

### 2.1. Study Area

A large amount of sub-fossil wood from the Holocene age is preserved in the alluvial sediments of the southern part of the Yamal Peninsula. Siberian larch (*Larix sibirica*) constitutes a vast majority of these samples (91%) and, to a lesser extent, Siberian spruce (*Picea obovata* Ledeb.) (6%) and downy birch (*Betula tortuosa* Ledeb.) (3%). Sub-fossil wood samples were collected from the valleys of the Khadyta, Yada-Yakha, Yuribey, and Tanlova rivers in an area located between 67°00′ and 67°50′ N and 68°30′ and 71°00′ E. The upstream parts of these rivers are forestless; sparse larch forests (birch–spruce–larch in the Khadyta River valley) occur mostly in the middle and lower parts of the rivers. Wood samples from live trees were collected from valleys of the same rivers (Figure 1).

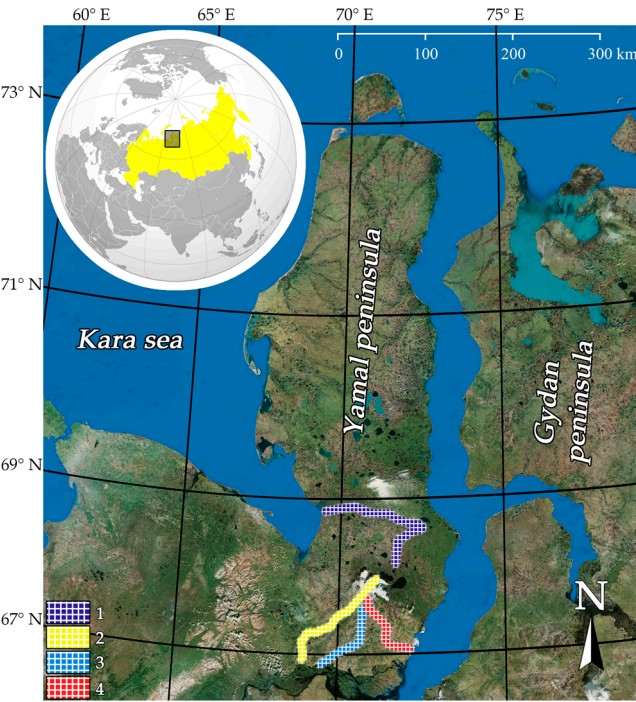

**Figure 1.** Research area map: 1—Yuribey, 2—Tanlova, 3—Khadyta, 4—Yada-Yakha.

### 2.2. Tree Ring Data

An analysis of anomalous structures was performed mainly on Siberian larch wood and, to a smaller extent, Siberian spruce due to low abundance. A total of 643 larch (of which 581 are sub-fossil) and 56 spruce (of which 36 are sub-fossil) samples were used to analyze anomalous structures, in which 81,649 and 9791 rings were examined, respectively. The distribution of samples over time is very heterogeneous, both in the number of samples and in the ratio of spruce and larch (Figure 2).

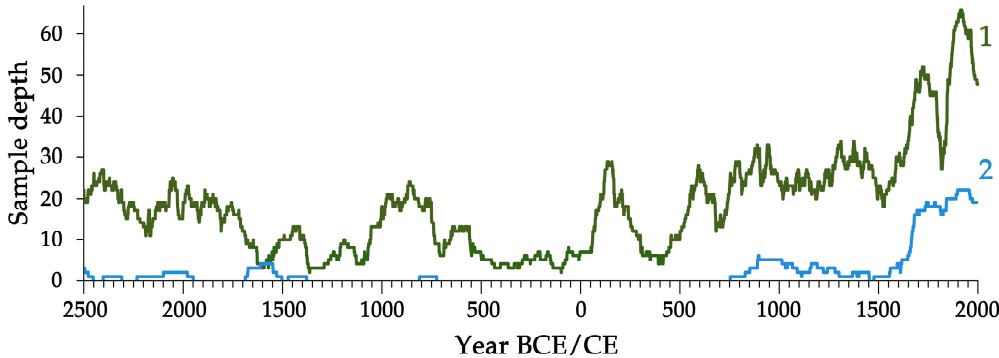

**Figure 2.** Depth of larch (1) and spruce (2) samples used for the analysis of anomalous structures.

### 2.3. Anomalous Structures in Tree Rings

To reconstruct extreme temperature events, we used three types of damage to the normal anatomical structure of annual rings of living and sub-fossil trees: frost injuries (frost rings), wood density fluctuations (false rings), and light rings (Figure 3). Abnormal structures were determined visually by examining cross-sections of wood samples with a binocular microscope at 32–56× magnification.

A typical frost ring consists of three zones [18]: a zone of deformed (curved) tracheids, a layer of an amorphous substance consisting of destroyed cells, and then a zone of abnormally sized and shaped tracheids (Figure 3A). It has been shown [13,19,20] that frost damage can also appear more subtle, sometimes as small as a change in the direction of radial growth of one or several rows of tracheids (Figure 3C). These are accompanied by more severe damage and are followed by rows of damaged (crumpled) cells and then rows of recovering tracheids with a changed direction of growth (Figure 3B). Finally, this damage may collectively look like a typical frost ring (Figure 3A). We considered all these changes in the normal structure as frost rings. As shown by observations [19,20] and experimental studies [18], frost rings are formed when the air temperature drops below zero during the period of cambium activity and xylem cell growth. Frost rings in larch wood on Yamal are formed when frosts occur from about June 20 to July 20 [21,22]. However, the sensitivity of trees to frost damage decreases as the bark thickness and tree diameter increase (e.g., with age). Therefore, frost injuries in larch and spruce are formed mainly in the central rings [20].

Wood-density fluctuation (false ring) is a layer of darker cells within the annual ring, which differs from neighboring layers in cell shape and size, as well as in the thickness of the cell wall (Figure 3D,E). Such a structure can occur when conditions during the growing season deteriorate for a relatively long time (one or two weeks), for example, a decrease in air temperature at the polar border of the forest, followed by normalized temperatures [19,23].

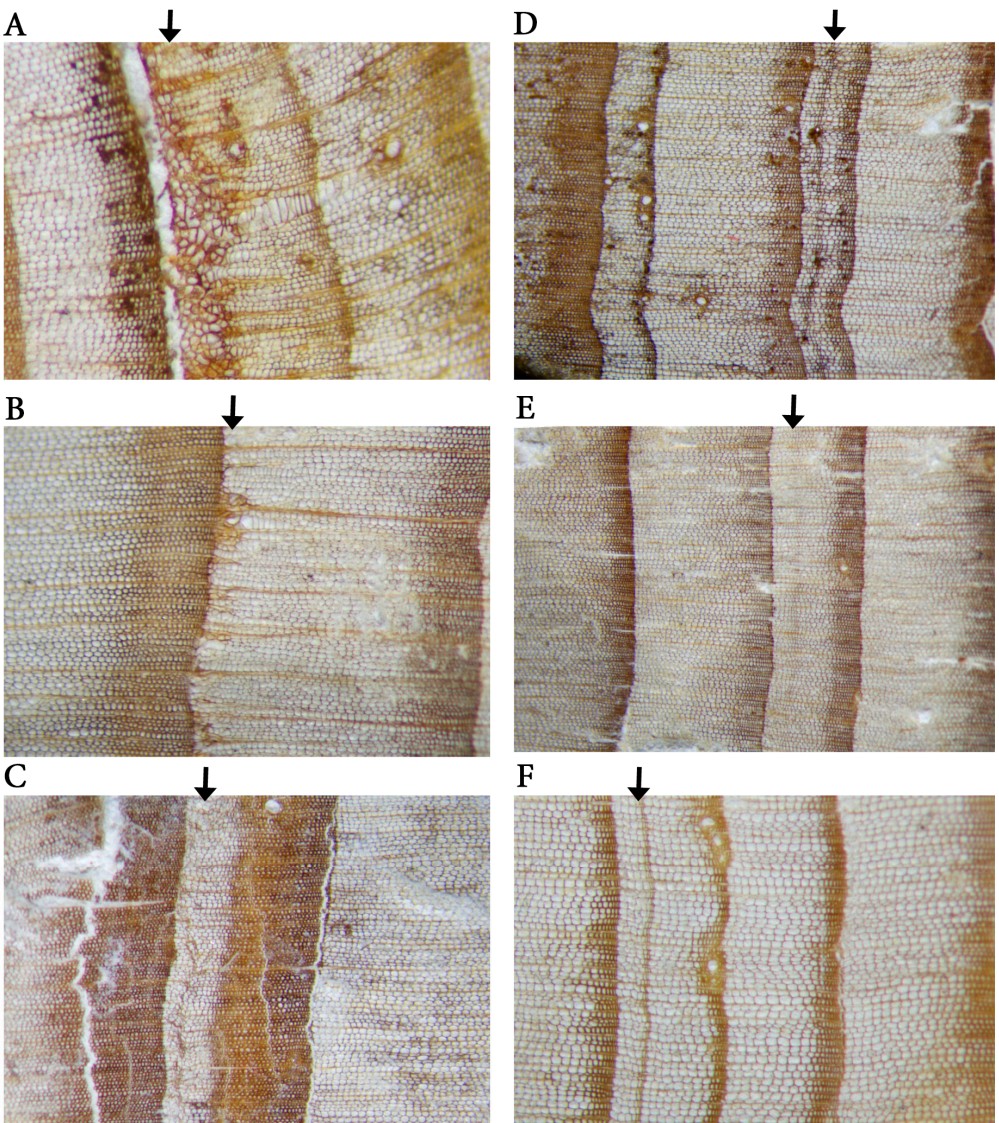

**Figure 3.** Types of anomalous structures in tree rings of larch used for reconstruction of extreme events: (**A**) typical frost ring, (**B**) less pronounced frost ring, (**C**) weakest degree of frost ring, (**D**) typical false ring, (**E**) less pronounced false ring, (**F**) light ring.

The light ring contains late-wood cells with thinner walls compared to normal cells (Figure 3F). Such rings form in years with unfavorable conditions. In the subarctic zone, light rings are formed in the cold (especially in July and August) and/or short summers [7,9,10,24,25]. In Yamal, the formation of light rings in larch is associated with the low average temperature in July-August, e.g., the second half of the growing season [13]. Light rings are conventionally called abnormal structures; however, they are formed quite often. In this study, we divided such rings into weakly and strongly expressed light rings and only considered the second type of light ring in the analysis.

In addition, "missing rings", e.g., the absence of wood growth on measured discs in any year, as well as extremely narrow rings (1% of narrowest rings according to [3]) were used to identify extreme summer seasons indicative of a very low average summer temperature.

Thus, based on the analysis of the aforementioned formations, it is possible to reconstruct climatic extremes such as summer frosts, sharp multiday decreases in air temperature during the vegetation period, and low average summer temperatures. To date, such reconstruction has been performed for the period from 2500 BCE to 2000 CE.

### 2.4. Statistical Methods

Analysis of the occurrence of abnormal structures showed that many years were marked with some anomalies. For example, in 615 of 4500 years (14%), at least one tree had a frost ring formed, and in 45% of years, at least one tree had a light ring. At the same time, in the majority of these cases, the disturbance of the normal structure of annual rings was observed only in single trees. This fact can be explained by two reasons. Firstly, microclimatic conditions are highly localized, thereby affecting the growth of individual trees. Secondly, in the case of frost rings, the earlier phenology of vegetation of some trees compared to others occurs and, therefore, there is a higher probability that these trees are exposed to frosts, which are a common event in Yamal at the very beginning of summer. To exclude this kind of "random" anomaly, we applied the following procedure. We assumed that any deviation from the normal structure of the annual ring could occur randomly and used our data to calculate the probability of occurrence of each structure in the annual ring. Then we estimated, using a Poisson distribution, the maximum number of anomalies that could appear in one of the 4500 studied years if they appeared at random. This calculation was done for all sample numbers ($n$) (Figure 2). In calculating each $n$, we made the simplified assumption that the number of samples was the same for the entire 4500-year period. When the number of detected anomalies exceeded the maximum possible number of random anomalies for a given replication, we considered that year to be "extreme".

### 2.5. The Calendar Used

It should be noted that in this work, the dates of formation of tree rings for the period BCE are given in the so-called "dendro-years", corresponding to the astronomical calendar. This system differs from the generally accepted system of chronology by the fact that for the convenience of data processing between the 1st year CE and the 1st year BCE, there is also a zero year. For example, the zero year in the "dendro-scale" corresponds to 1 year BCE in the Gregorian system of time calculation, and then all the years BCE are shifted one year ahead (e.g., all ages are reported using the ISO 8601 international standard).

## 3. Results

### 3.1. Changes in the Frequency of Anomalous Structures over Time

In the analyzed larch tree rings on Yamal, 1.0% were frost-damaged and 0.38% of the examined annual rings were false; in spruce, 1.2% were damaged and 0.12% were false. The low frequency of these anomalies is explained by the fact that under the conditions of a very short (a few weeks) growing season near the polar forest boundary, it is unlikely that normal growth will be restored after a long cold period. In the analyzed larch tree rings in Yamal, light rings accounted for 11% of all annual rings, and in spruces this value was 10%. Only 0.17% of the rings were recorded as missing in larch and 0.06% in spruce.

The number of identified extreme years for individual periods in our methodology depends on the sample abundance of these periods. For time intervals with a low number of samples, only the most extreme years will be reliably identified. Therefore, it would be more correct to estimate the dynamics of the frequency of extreme events with significant fluctuations of sample depth (Figure 2) in an indirect way, namely, based on the percentage of anomalous rings in all analyzed tree rings for some multiyear time interval (without reference to individual years).

Figure 4 shows the changes in the percentage of anomalous rings of larches over the last 4500 years over 50 year intervals. For frost and false rings, pronounced millennial trends are noted. Up to 500 BCE, the frequency of frost rings was relatively low (0.4% on average) and then significantly increased up to 1.3% on average. In contrast, for false rings, in the first half of the reconstruction up to 100 BCE, the frequency was relatively high (0.57% on average) compared to the following two millennia (0.27% on average). For narrow and light rings, no noticeable trends were revealed.

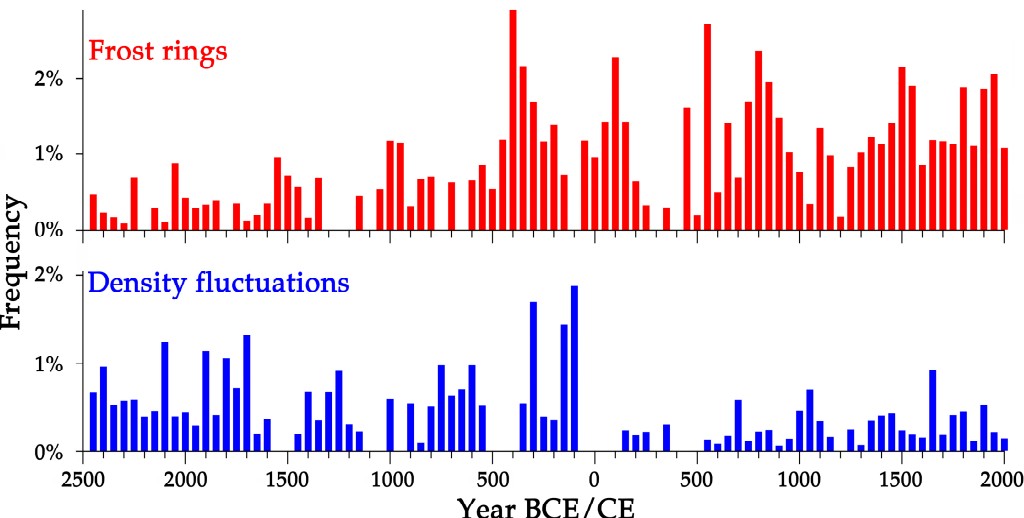

**Figure 4.** Frequency of anomalies in larch tree rings per 50-year intervals.

### 3.2. Dates of Extreme Events

The results of the reconstruction of extreme summer seasons are presented in Table 1. Extremum intensity was determined subjectively, based on data on the percentage of samples with ring anomalies and, in the case of frost and false rings, on the degree of severity of annual tree ring anomalies.

**Table 1.** List of extreme years according to the analysis of anomalous anatomical structures in Siberian larch and Siberian spruce wood.

| Year | I | A | Year | I | A | Year | I | A | Year | I | A | Year | I | A |
|---|---|---|---|---|---|---|---|---|---|---|---|---|---|---|
| | BCE | | 1386 | * | $ll_pM$ | 78 | * | f | 976 | * | $ll_p$ | 1538 | * | $fll_p$ |
| 2473 | * | fld | 1172 | * | lN | 46 | * | n | 985 | * | f | 1548 | * | f |
| 2450 | * | l | 1169 | * | L | 42 | * | fl | 1002 | * | l | 1560 | * | $fll_p$ |
| 2422 | * | fl | 1127 | * | lN | | CE | | 1003 | * | l | 1569 | * | $d_p$ |
| 2397 | * | l | 1122 | * | L | 49 | * | n | 1004 | * | ln | 1576 | * | m |
| 2378 | * | ld | 1011 | * | ld | 110 | * | f | 1012 | * | d | 1585 | * | $fll_p$ |
| 2256 | * | fl | 989 | * | l | 134 | * | flm | 1025 | * | $fl_p$ | 1601 | *** | $FF_pl_p$ |
| 2251 | * | fl | 988 | * | n | 143 | ** | Fmn | 1029 | * | l | 1609 | * | d |
| 2244 | * | l | 982 | ** | F | 155 | * | fl | 1040 | * | $ll_p$ | 1617 | * | $ll_p$ |
| 2053 | *** | $FLd_p$ | 975 | * | l | 180 | * | l | 1051 | * | $fLl_p$ | 1628 | * | $l_p$ |
| 2035 | * | $Ll_p$ | 965 | * | mn | 194 | * | f | 1055 | * | l | 1629 | * | $l_p$ |
| 1996 | * | $ll_p$ | 919 | ** | F | 404 | ** | F | 1077 | * | f | 1631 | * | $ld_p$ |
| 1973 | * | l | 904 | * | l | 536 | * | f | 1099 | * | $ll_p$ | 1634 | * | $fll_p$ |
| 1972 | * | $ll_p$ | 883 | ** | lMN | 537 | * | f | 1109 | * | fl | 1642 | * | m |
| 1966 | * | $ll_p$ | 866 | * | l | 543 | ** | F | 1122 | * | $Fl_p$ | 1666 | * | f |
| 1947 | * | ld | 861 | * | flM | 545 | * | f | 1133 | * | $ff_p$ | 1677 | * | l |
| 1935 | ** | LMn | 828 | * | fl | 546 | * | n | 1136 | * | $fl_p$ | 1679 | * | $ff_pll_p$ |
| 1920 | * | Ln | 823 | * | l | 596 | * | l | 1151 | * | $ll_p$ | 1683 | * | $ff_pll_p$ |
| 1918 | * | l | 809 | * | $fll_p$ | 623 | * | lmn | 1170 | * | lM | 1694 | * | d |
| 1812 | * | d | 800 | * | $fl_pd$ | 627 | * | F | 1172 | * | $fLl_p$ | 1699 | * | $ll_p$ |
| 1728 | * | d | 792 | * | $ll_p$ | 637 | * | Mn | 1201 | * | $fll_p$ | 1714 | * | fd |
| 1726 | * | fl | 741 | * | $fll_p$ | 639 | * | fmn | 1209 | ** | $Fl_p$ | 1717 | * | $ll_p$ |
| 1655 | * | $F_pl$ | 717 | * | L | 640 | ** | lmN | 1217 | * | f | 1723 | * | l |
| 1650 | * | l | 680 | * | L | 652 | * | f | 1234 | * | $l_p$ | 1730 | * | $fll_p$ |
| 1647 | *** | lMn | 658 | * | L | 684 | * | l | 1236 | * | $l_p$ | 1732 | * | l |
| 1634 | * | $ll_p$ | 609 | * | fLmn | 738 | * | F | 1259 | * | $ff_pl$ | 1745 | * | $fll_p$ |

**Table 1.** *Cont.*

| Year | I | A | Year | I | A | Year | I | A | Year | I | A | Year | I | A |
|---|---|---|---|---|---|---|---|---|---|---|---|---|---|---|
| 1626 | ★★★ | $F_pLMN$ | 593 | ★ | mN | 754 | ★ | $Fl_p$ | 1270 | ★ | $fl_p$ | 1757 | ★ | $l_p$ |
| 1625 | ★ | n | 569 | ★ | Ln | 757 | ★ | $fl_p$ | 1278 | ★ | $fll_p$ | 1776 | ★ | fd |
| 1622 | ★ | $f_pl$ | 550 | ★ | lMN | 779 | ★ | F | 1288 | ★ | F | 1783 | ★ | F |
| 1617 | ★ | $Ll_pn$ | 489 | ★ | Ln | 784 | ★ | $fl_p$ | 1300 | ★ | l | 1816 | ★ | MN |
| 1569 | ★ | $ll_pM$ | 472 | ★ | fl | 791 | ★ | f | 1312 | ★ | $fl_pd$ | 1818 | ★★ | $MM_pN$ |
| 1557 | ★ | $fl_p$ | 464 | ★ | fl | 800 | ★ | $l_p$ | 1342 | ★ | $fll_pn$ | 1820 | ★ | Mn |
| 1553 | ★★★ | $Ll_pMn$ | 420 | ★ | flN | 801 | ★ | $FF_p$ | 1347 | ★ | dMn | 1862 | ★ | $ff_pl_p$ |
| 1538 | ★★ | $Ll_pM$ | 415 | ★ | N | 814 | ★ | M | 1350 | ★ | $f_pl$ | 1866 | ★ | $fl_p$ |
| 1523 | ★ | $ll_p$ | 402 | ★ | F | 816 | ★ | Mn | 1352 | ★ | $fl_p$ | 1867 | ★ | d |
| 1507 | ★ | f | 362 | ★ | n | 818 | ★ | M | 1372 | ★ | fl | 1872 | ★ | $ff_p$ |
| 1493 | ★ | f | 358 | ★ | F | 857 | ★ | $f_p$ | 1383 | ★ | $fll_pn$ | 1879 | ★ | $ff_p$ |
| 1492 | ★ | ln | 345 | ★ | LN | 878 | ★ | $ff_p$ | 1391 | ★ | $fl_p$ | 1884 | ★ | $f_p$ |
| 1486 | ★ | f | 339 | ★ | f | 881 | ★ | l | 1435 | ★ | $fl_p$ | 1912 | ★ | $ff_pl_p$ |
| 1465 | ★ | l | 338 | ★ | d | 884 | ★ | fl | 1440 | ★★ | $Fl_p$ | 1958 | ★ | $ff_p$ |
| 1462 | ★ | $ll_p$ | 321 | ★ | lN | 901 | ★ | f | 1453 | ★★ | fMn | 1970 | ★ | $f_p$ |
| 1443 | ★ | $ll_pd$ | 318 | ★ | n | 903 | ★ | $fll_pm$ | 1466 | ★★★ | F | 1978 | ★ | $ff_p$ |
| 1417 | ★ | l | 250 | ★ | f | 912 | ★ | fm | 1481 | ★★ | $fF_pl$ | 1980 | ★ | l |
| 1410 | ★★ | $l_pM$ | 248 | ★ | N | 927 | ★ | F | 1509 | ★ | $ll_pd$ | | | |
| 1403 | ★ | L | 242 | ★ | f | 940 | ★ | $ll_p$ | 1527 | ★ | $fl_p$ | | | |
| 1401 | ★★ | $Ll_pMN$ | 241 | ★ | n | 963 | ★ | fl | 1529 | ★ | $l_pmN$ | | | |
| 1394 | ★ | Ln | 154 | ★ | d | 972 | ★ | $f_pLl_p$ | 1531 | ★ | l | | | |

I—extremum intensity (★ moderate, ★★ strong, ★★★ maximal); A—types of anomalies: f—frost; l—light; d—false; m—missed; n—narrow rings. Capital letters denote a high proportion of samples with a given anomaly. Designations with the index "p" refer to anomalies in spruce, without the index—in larch.

A total of 229 extreme years were identified. The most exceptional extreme events were in 2053, 1647, 1626, 1553 BCE and 1466, 1601 CE. The years 1935, 1538, 1410, 1401, 982, 919, 883 BCE, 143, 404, 543, 640, 1209, 1440, 1453, 1481, and 1818 CE are also worthy of mention.

Detailed analysis of Table 1 reveals an irregular distribution of extreme years. At the century scale, the incomplete identification of extremum can be partially explained by the low availability of samples for some periods. However, the uneven distribution of extreme years is also noticeable in shorter time intervals. In 1002–1004 CE there was a series of three consecutive extreme summers. Similar series, three extrema in 4 years, took place in 637–640 CE and 1628–1631 CE, three extrema in 5 years in 814–818 CE, 1527–1531 CE, 1816–1820 CE, and in the last case, their intensity was the highest. Other sequences of extrema occurred from 536–546 CE, as well as from 1628–1642 CE, a part of which was the four years mentioned above. In turn, this interval could be considered part of the anomalous period 1601–1642 CE (eight extrema), or 1601–1699 CE (14 extrema), or even 1527–1732 CE (28 extrema in 206 years). Perhaps, our data are another confirmation of the fact that adverse conditions during the Little Ice Age were caused not only by low average temperatures but also by a high frequency of extreme temperature events during the growing season.

The periods 1417–1386 BCE (six extrema in 32 years), 1862–1884 CE (six extrema in 23 years), 779–818 CE (eight extrema in 40 years), 963–1055 CE (13 extrema in 93 years) are also worth mentioning. Finally, the period from 1655 to 1617 BCE stands out. Although the frequency of extreme years during this period (seven extrema during 39 years) was not as high as during most of the periods listed above, two exceptionally anomalous summer seasons (1647 and 1626 BCE) fell within this interval. Therefore, it was probably the most unfavorable time for tree growth during the last 4500 years.

## 4. Discussion

### 4.1. Changes in the Frequency of Frost Rings and Density Fluctuations over Time

The observed differences in the time distribution of frost rings and false rings (Figure 3) indicate, in our opinion, differences in the length of the growing season and average temperature. The formation of false rings, as mentioned above, requires a relatively long period of several cold days followed by a number of days of favourable growth conditions, e.g., a total of at least 2–3 weeks during the growing season. Taking into account that the period of tree-ring formation in Yamal is only a few weeks long ([17]), the probability of false-ring formation is higher in years when the growing season is longer. Most likely, in the period from 4.5 to 2.1 thousand years ago, the growing season indeed lasted longer, so the probability of the formation of density fluctuations under appropriate conditions was higher. The gradual decrease in temperature over seven thousand years in Yamal has been shown in an earlier study [3]. Therefore, it is reasonable to assume that the probability of frost in colder times was higher, so the proportion of frost rings increased. Thus, it can be assumed that the opposite changes observed in the frequency of false- and frost-ring formation reflect changes resulting from a shorter and cooler growing season.

### 4.2. The Spatial Scale of Extreme Events

Anomalous structures in tree rings may reflect extreme events on both regional and wider spatial scales. To assess this scale, we compared data for Yamal with similar tree-ring data in other regions. In cases where an extreme event in Yamal coincided with an extreme event in a remote region, we assumed that the most likely cause of this coincidence could be a large volcanic eruption that led to an extremum over a large area. To test this assumption, we used data that examined traces of large volcanic eruptions in the ice cores of Greenland and Antarctica. Because a postvolcanic climatic extreme can happen in the 1–3 years following an eruption and can occur in different areas with a slight lag, we considered it a coincidence when anomalies were recorded in the tree rings of remote areas in the same year as in Yamal, or slightly earlier or later, and traces in ice cores were recorded in the same or the previous 1–3 years. We have also, wherever possible, used historical information.

In 1647 BCE, one of the most extreme years in Yamal, cold summers were also recorded in Finland [26] and one year prior in the southwestern United States [15]. Traces of a volcanic eruption dating from approximately the same year (1643 BCE) were found in Greenland in Dye column three [27]. According to Pearce et al. (2003) and Keenan et al. (2004), it is shown that these are traces of the eruption of the Anyakchak volcano in Alaska [28,29].

The extreme event of 1626 BCE, one of the most significant ones in Yamal, is well-known in other areas as well [15,30,31]. Traces of a large eruption at approximately the same time were also found in the Greenland ice cores [27,32], so there is no doubt that the extreme event of 1626 BCE was caused by a volcanic eruption. Scientific discussions in recent decades have debated the source of this eruption and one hypothesis is that the source was the eruption of the Santorini volcano (Thera) in the Aegean Sea that ended the Minoan civilization [2,31].

The absence of coincidence of extreme years during more than eight centuries (from 1416 to 551 BCE) may be due to the relatively low number of occurrences of extrema in the Yamal during this period. However, there was also a decrease in the number of anomalous rings in North America during the same period (except for the twelfth century BCE) [15].

The extreme event of 42 BCE was marked in Yamal by anomalies of tree rings. Frost rings formed in the same year in the annual rings of pines growing in the United States [15], and in the territory of Finland [26], the trees had very little summer growth. Glacier layers were dated to around the same year in the GISP2 and GRIP, and Dye ice cores showed evidence of a volcanic eruption [27,32]. There are also extant written accounts of unusual phenomena in 44 BCE (43 dendro-year BCE) and the following two years [33]. Plutarch (about 150 years after the event) portrayed the state of the "dim sun" after the assassination of Julius Caesar (15 March 44 BCE) as follows: "Throughout that year its disk rose pale

and without radiance, and the heat from it was weak and barren, so that the air . . . was dark and heavy because of the impotence of the heat that permeated it, and the fruit, flawed and half-ripe, wilted and dried because of the coldness of the atmosphere." Ovid (about 10 CE) called that year's moon "bloody" and Venus "dark red." The Roman poet Calpurnius (about 69 CE) mentioned a "blood-colored" comet observed in 44 BCE. The red tones of the observed objects are associated with atmospheric haze. According to the Han dynasty chronicles in April-May 43 BCE (42 dendro-year BCE), late snow fell in China. A late frost killed the mulberry trees, and in May-June, the sun was bluish-white and did not create shadows; at noon there were shadows, but vague. In October, a frost ruined the crops and a famine ensued. Some researchers believe that the effects observed were caused by the eruption of the Mt. Etna volcano in 44 BCE. [34]. However, attributing this eruption to the Etna volcano may simply be a random coincidence because there is so much historical evidence about it. Therefore, it is likely that a larger explosive eruption of the volcano in another area occurred at the same time. For example, we know about a very large eruption of Ambrym volcano (Vanuatu) around 50 BCE (volcanic deposits were dated by radiocarbon method) [35].

Furthermore, for almost six centuries there was only one coincidence in the dates of extreme events in Yamal and those occurring in other regions, which is again associated with a low number of extreme years in Yamal.

The summer of 180 CE was extreme in Yamal, the United States [15], and Taimyr [36]. There are also traces of volcanic eruptions in both Greenlandic and Antarctic ice cores [37,38]. In the Antarctic, this signal is one of the most prominent in the last four thousand years. A very large eruption of the Taupo volcano in New Zealand was dated to approximately the same year [39]. However, later clarification of radiocarbon dating has rejuvenated this event by about 50 years [40].

The next interesting event on a global scale, which also left a trace in the larch tree rings in Yamal, was the so-called "mysterious cloud" of 536. Byzantine sources describe an unusual fog that covered a vast area in 536–537 CE: "the sun was dark and its darkness lasted for eighteen months; each day it shone for about four hours, and still this light was only a feeble shadow . . . , the sun had an unusual bluish color, . . . the fruits did not ripen . . . ". Cold and drought led to crop failures in Italy and Mesopotamia, causing famines in subsequent years. A similar phenomenon is recorded in Chinese sources. In many provinces of China in July and August of 536, there were frosts and snow that killed crops and caused famine among the population, which lasted until 538 (historical information cited from [41]). There has long been a debate in the academic community about the cause of the climatic cataclysms that began in 536. The most plausible assumption was that it could be attributed to a large volcanic eruption. However, for a long time, there was no evidence of a suitable eruption in the ice cores from Greenland and Antarctica to explain it. This was due to the sparse data on the content of sulfates, which are markers of volcanic eruptions, and gaps in the dating of ice layers. Such a gap in the data led to many assumptions about the extraterrestrial origin of the mysterious phenomenon. Publications were linking it to the fall to Earth of a large comet, an asteroid, or the passage of the solar system through a cloud of cosmic dust.

In 2008, an international team of glacier researchers and dendrochronologists summarized data on the width of annual tree rings and ice cores. The dendrochronologists found that the cooling, which began in 536 CE, covered all of Eurasia and lasted for a long time of about 15 years. [42]. That is rather unusual because even very large volcanic eruptions only reduce the temperature by 3–5 years. The other part of the group, glaciologists, as a result of large-scale sampling and highly accurate determination of sulfates in layers of ice cores from three areas of Greenland, revealed traces of a large volcanic eruption in the ice layers dated to $534 \pm 2$ years CE. The mass of gases released into the atmosphere by this eruption, as estimated by sulfate accumulation in the ice columns, was 40% higher than the 1815 eruption of the Tambora volcano—the largest eruption in the most recent 200 years. In other words, this volcanic eruption was one of the most powerful of the last

two millennia. There was also a trace of another eruption, a local, Icelandic volcano, dated four years earlier, which glaciologists consider to have caused no climatic effect. However, the dendrochronological method was later used to correct an error in the dating of the ice layers for several years [43]. It was determined that the local volcano was the sole source of the 536 eruptions from the ice core and dendrochronological records, whose dry tropospheric fog covered Eurasia that same year. A very powerful eruption occurred about four years later and caused a climatic response due to the formation of a layer of aerosol in the stratosphere. Therefore, its consequences began to be felt in different regions of the Earth with a delay of 1–4 years. Further studies showed that the most likely source of the 540 eruption was the Ilopango volcano in Central America [44]. Such a double volcanic strike led to an unusually prolonged cooling period [45]. The anomaly in the tree rings of 536 coincides in all regions on Earth, whereas the anomaly of the early 540s had a difference of 1–2 years, depending on the region. In Taimyr and Mongolia, similar to Yamal, the extreme event occurred in 543, but in Fennoscandia, the event occurred in 542–543, in the USA it occurred in 541–543, and in Yakutia it occurred in 540–541 [46].

The event of 627, noted in the tree rings in Yamal, North America [15], in the north of Central [36,47] and Eastern Siberia (and in Fennoscandia in 628 [26]) is also associated with a major volcanic eruption. Traces of a major volcanic eruption have been found in a layer of Greenlandic ice dating back to 626 [43]. From the European chronicles, it is known that starting from October 626, a dry fog covered a vast territory during 8–9 months from Ireland to the eastern Mediterranean [48]. Records of a sudden and huge disaster in the Eastern Turkic Empire and subsequent great famine and epidemics in the years 627–629 are in the historical Chinese literature [49,50]. They mention frost in the midsummer and early autumn, heavy snowfalls and deep snow cover, which led to the death of all sheep and horses, followed by multi-year starvation of the nomadic Türks, high mortality, and eventually the collapse of the mighty Turkic Empire. In China itself (Tang Dynasty) a frost in the late summer of 627 also occurred, which destroyed crops in several provinces. The same events were noted in 628–629.

Regarding the event around 940, there was a long debate about the exact date of the Eldgjá volcano eruption [37,51]. The attribution of this event to Eldgjá was assumed correctly by Sigl et al. [43] and Oppenheimer et al. [52]. The Eldgjá lava flood is considered as Iceland's largest volcanic eruption of the Common Era [53]. Recent tree-ring-based reconstructions reveal pronounced Northern Hemisphere summer cooling in 940 CE [36,54], which is consistent with the eruption's high emission of sulfur into the atmosphere.

Further, the coincidence of extreme years in Yamal and other regions is quite frequent. This is due both to the high number of extreme years identified in Yamal and to more detailed information on anomalous rings from other regions.

A remarkable extreme, one that had a global scale, left traces on Yamal in the annual rings formed in 1259. In 1258 and 1259 in Mongolia, frost rings were also formed in Siberian cedar tree rings [55]. In addition, in 1258 in eastern Canada, a significant proportion of spruce trees formed light rings [54]. A significant decrease in growth in 1258–1259 was observed in larches in the Polar Urals [13], Taimyr [36], and Yakutia [56]. This was a consequence of a large volcanic eruption that left its mark on the polar ice cores of both the Northern and Southern Hemispheres [27,37,38,57]. According to some estimates, the emission of gases from this volcano was twice as large as that from the eruption of Tambora in 1815. That is, it was the largest eruption in the world in the last thousand years, maybe even 7000 years [32], which began in the middle of 1257 [58]. The source of this eruption has since been identified as the Samalas volcano [59]. According to historical data [60], the climatic and demographic consequences of this volcanic eruption were not particularly impressive. In 1258, a dense layer of clouds covered the sky in France throughout the summer, so it was unclear whether it was summer or autumn. At the time of the lunar eclipse of 18 May 1258, in England, one observed not the usual change in the color of the moon (reddening), but its complete disappearance, apparently due to the presence of dense clouds in the stratosphere. The summer in northern Italy and western Germany was also

cold and rainy and led to crop failure and famine. There was also famine in Iraq, Syria, and southeastern Turkey. In both Europe and the Middle East, however, it could have been caused by political reasons as well. In Russia, weather anomalies (frosts) were observed in the summer of 1259 [61].

The damages of 1453 are noted almost everywhere [15,54,62,63]. In addition, there is historical information about cold rainy summers that year and in the following 4 years in Russia, as well as early autumn frosts in 1453 [61]. It is believed that the cause of the global temperature drop in the summer of 1453 was the eruption of the Kuwae volcano (Vanuatu) [64].

The summer season of 1466 was perhaps the most extreme in Yamal and the Northern Hemisphere [43]. Some chronicles contain information [61] that snow covered the land as late as May 26 in Russia that year. On August 18, there was already a frost and the following winter and summer were cold. The event of this year was also reflected in the annual rings of trees from North America [15], Mongolia [63], and the Polar Urals [19]. The dating of volcanic traces in the GISP2 ice core in Greenland [37] is closer to 1466 than to 1453. Ice cores from some other Greenland glaciers have a double peak of volcanic eruption traces. The first of them coincides in the date identified in the Antarctic cores, and the second is in the early 1460s (there is no such peak in Antarctica). Therefore, it is believed [65] that the signal of the 1460s refers to the eruption of a volcano located near Greenland, possibly in Iceland.

In 1601, anomalous tree rings formed in many regions of the world. Unusual natural phenomena that summer were observed in Europe. In England, "June was very cold, with frosts every morning" [33]. In the European part of Russia in 1601, "early in the summer, there were great frosts" and crops and "all vegetables" died. Within three years of the beginning of the famine in Moscow, 120 thousand people died [61]. According to tree ring density reconstruction of summer temperatures of the Northern Hemisphere [64], the summer of 1601 was the coldest in the last 600 years. It is interesting to note that, according to reconstructions of summer temperatures based on the ring width of trees and shrubs from the Polar Urals and Yamal [13], the average summer temperature of 1601 was at the level of the long-term average. Nevertheless, some global events left their mark on tree rings in this area. Studies [66] found that this cooling was caused by the eruption of the Huaynaputina volcano in Peru in February-March 1600, which represents the world's largest eruption in the last 500 years.

The next noteworthy year was 1783. A sharp drop in air temperature that summer, which has already been accurately established by many researchers [67,68], was caused by dry fog, which, according to historical data, covered an area from England to the Altai region. Unusually cold weather also came to Alaska [69], where rivers and lakes were covered with ice in midsummer. This anomaly was caused by the eruption of the basaltic fissure eruption of Laki volcano, which began on the 8th of June 1783 and was one of the largest, in terms of the mass of $SO_2$ emitted, high-latitude eruptions of the last millennium [43].

The extreme event of 1816 is well-known from numerous historical documents in Europe and North America as the "year without summer". It is also well-known that the cause of this cooling was the eruption of the Tambora volcano in April 1815, one of the largest in recent centuries [70]. Perhaps the extremes of 1818 and 1820 were also a more distant consequence of the same eruption.

The formation of extreme tree rings in many areas of the world in 1884 was most likely caused by the eruption of the Krakatoa volcano in Indonesia which led to a significant decrease in sunshine duration in Europe for several years after the eruption [71]. Another abnormal tree ring on Yamal was formed in 1912, as well as in North America [15,54,62]. The cause of the formation of the light ring was the eruption of the Katmai (Novarupta) volcano in 1912 in Alaska, which was the largest eruption in the world in the twentieth century and produced the largest ash-flow sheet in the historical record [72].

A summary of the discussion is shown in Figure 5. Tree-ring anomalies in Yamal reflect global extreme climate events caused by major volcanic eruptions. However, it is necessary to take into account that the intensity of the anomalous structures does not always correlate with the strength of the eruption.

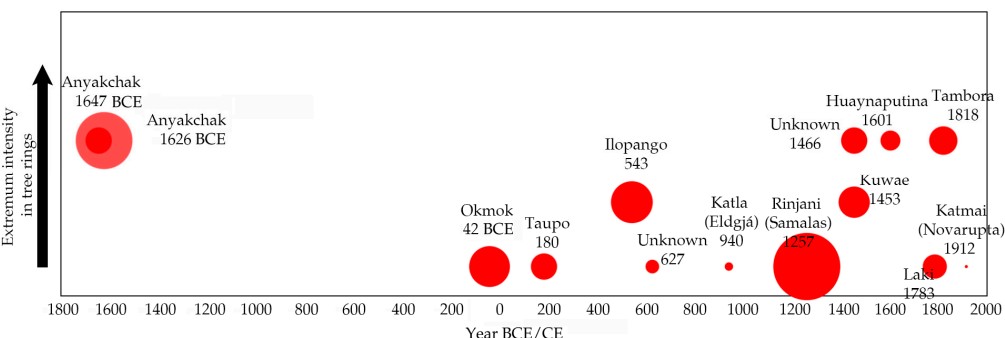

**Figure 5.** Years of global extreme events with "footprints" observed in Yamal tree rings. The size of the marker reflects the strength of the volcanic eruption, and the position of the marker relative to the *x*-axis indicates the intensity of the anomalies in the tree rings.

## 5. Conclusions

Analysis of anomalous structures in tree rings from the Yamal peninsula in the north of Western Siberia allowed us to identify years when there was an extreme decrease in air temperature of different durations during the summer season. A total of 229 extreme years were identified. The most significant temperature extremes were noted in 2053, 1935, 1647, 1626, 1553, 1538, 1410, 1401, 982, 919, 883 BCE, 143, 404, 543, 640, 1209, 1440, 1453, 1466, 1481, 1601 and 1818 CE.

Comparing our results with the data on the dates of formation of anomalous tree rings in other regions of the world showed that, in some cases, the extrema on the Yamal peninsula coincided with similar extrema in other remote regions on Earth (e.g., all probability had occurred on a global scale). In most cases, when the dates of extrema coincided in different regions, there were also traces of large volcanic eruptions in the ice cores of Greenland and Antarctica dated to approximately the same year. Thus, the cause of the extreme summer cooling, of a global scale in most cases, can be considered a consequence of large volcanic eruptions.

**Author Contributions:** Conceptualization, R.H.; Methodology, R.H. and L.G.; Formal Analysis, R.H., L.G. and I.H.; Investigation, L.G. and I.H.; Resources, V.K. and R.H.; Data Curation, L.G.; Writing—Original Draft Preparation, V.K. and R.H.; Writing—Review & Editing, V.K. and R.H.; Visualization, V.B., V.K. and R.H. All authors have read and agreed to the published version of the manuscript.

**Funding:** The research funding from the Ministry of Science and Higher Education of the Russian Federation (Ural Federal University Program of Development within the Priority-2030 Program) is gratefully acknowledged.

**Data Availability Statement:** The data that support the findings of this study are not openly available and are available from the corresponding author upon reasonable request.

**Conflicts of Interest:** The authors declare no conflict of interest. The funders had no role in the design of the study; in the collection, analyses, or interpretation of data; in the writing of the manuscript; or in the decision to publish the results.

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
