# Peer review of "A 4500-Year Tree-Ring Record of Extreme Climatic Events on the Yamal Peninsula"

_forests, doi:10.3390/f14030574_

Round 1
Reviewer 1 Report
Review comments for Forests-2236328:
The authors evaluated the extreme climatic events in Yamal over the past 4500-year using the frost rings, false rings and light rings from a previously dated and published tree-ring data. It is significance to study the climatic extremes on a long-term time scale, which will help us accurately evaluate the current and future climate change. This manuscript mainly includes two parts: first, identify the extreme events over the past 4500-year; second, discuss and verify part of the extreme events by comparing with other paleoclimate records. The research method and results are concise. Here are some comments and suggestions for this manuscript.
Main concerns:
1. For a science paper, the introduction of this manuscript is not sufficient exposition. The current research progress about the extreme climatic events, especially for the long-term studies using the tree-ring data and other paleoclimate records should be reviewed. Secondly, the innovation, study aims and main contents or organization of this manuscript should be stated clearly in the introduction.
2. The abstract gives the readers a misleading impression that all the 229 extreme events were caused by the volcanic eruptions, but actually only few of the extreme events were verified in the discussion.
3. There are some contradictions in the discussion. For example, in line 220-238, the extreme event was marked by tree-ring in 42 BCE, however, the volcanic eruption was in 44 BCE, later than the abnormal rings were formed. The volcanic eruptions in 1258 (“was the largest eruption in the world in the last thousand years, maybe even 7000 years”), in 536 CE (“was 40% higher than the 1815 eruption of the Tambora volcano- the largest eruption in the most recent 200 years), and some other historical volcanic eruptions were regarded as the strongest. However, the abnormal anatomical structures in tree rings were not the most significant according to the Table 1. Why?
4. There is only one subheading in the discussion, and the presentation of discussion is disorder. Some of the extreme events are global and hemispherical scale, while some are local scale, which should be described separately. In addition, some paragraph (for example the line 309-314) is too short to explain one extreme event, while some paragraphs (for example the line 253-296) are too long. The logical structure of discussion should be reorganized.
5. Most of the discussion focus on the verifications of tree-ring recorded extreme events resulted by volcanic eruptions using ice layers studies, document literature, and so on. Are there some new discovery reflected by this 4500-year record of extreme climatic events?
Revise suggestions:
1. In line 297-308, there is no information about the volcanic eruption.
2. In line 253-254, for the 536 CE event, does only the larch tree rings record this extreme event?
3. Materials and Methods, suggest using the subheading to sort the contents, such as: 2.1 study area; 2.2 tree-ring data; 2.3 abnormal anatomical structures in tree rings; 2.4 statistical method. Also, some information belongs to the results, such as the line 89-90, should be moved to the Results part.
4. Please add one figure to clarify the study area, which will be more helpful than the text in the line 50-52.
5. Please summarize the main information of discussion into one plot. That will be visual.
Author Response
We are grateful to the reviewer for your comments and advice on improving the manuscript. We tried to take into account all the recommendations made in the review. The corrected version of the manuscript passed the proofreading procedure and the numbering of figures has also changed. We’ve highlighted the corrected fragments of the manuscript in color.
Please see the attachment

Reviewer 2 Report
The authors analyzed the history of past extreme climate events by counting the frequencies of several kinds of abnormal tree rings in larch and spruce on the Yamal Peninsula over the past 4500 years. Under the background of frequent extreme climate, this research has certain scientific significance. The MS is well written and, on the whole, meets the standards for publication. I personally have the following minor comments.
1) After calculated the relative frequencies of the two kinds of abnormal tree rings, you did not analyze it with climate, but concluded the relevant types of extreme climate events based on previous experience, and then verified them to some extent in the discussion. Can extreme weather events in the measurement phase with abnormal tree ring frequencies be analyzed and verified? This can more confirm the accuracy of the results.
2) L156-157, "This is probably because, in the period from 4.5 to 2.1 thousand years age..." That's more of a discussion than result.
3) Figure3 shows the abnormal characteristic frequency of larch. Thus the tree species should be reflected in its legend.
4) 164-166. How are extreme events identified? Could you be more specific?
Author Response

(The authors gave the same response as above.)

Reviewer 3 Report
Dear authors,
Please find my comments in the attached file.
Regards

Author Response

(The authors gave the same response as above.)

Round 2
Reviewer 3 Report
Dear Authors,
Thank you for taking my comments into account.
Regards
Author Response
We are grateful to the reviewer for his comments and advice on improving the manuscript.